# Novel αO-conotoxin GeXIVA[1,2] Nonaddictive Analgesic with Pharmacokinetic Modelling-Based Mechanistic Assessment

**DOI:** 10.3390/pharmaceutics14091789

**Published:** 2022-08-26

**Authors:** Xiaoyu Zhu, Mei Yuan, Huanbai Wang, Dongting Zhangsun, Gang Yu, Jinjing Che, Sulan Luo

**Affiliations:** 1Key Laboratory of Tropical Biological Resources, Ministry of Education, Hainan University, Haikou 570228, China; 2State Key Laboratory of Toxicology and Medical Countermeasures, Beijing Institute of Pharmacology and Toxicology, Beijing 100850, China; 3Medical School, Guangxi University, Nanning 530004, China

**Keywords:** αO-conotoxin GeXIVA[1,2], nonaddictive analgesic, α9α10 nAChR inhibitor, pharmacokinetics, PopPK and PK-PD modelling, mechanism

## Abstract

αO-conotoxin GeXIVA[1,2] was isolated in our laboratory from *Conus generalis,* a snail native to the South China Sea, and is a novel, nonaddictive, intramuscularly administered analgesic targeting the α9α10 nicotinic acetylcholine receptor (nAChR) with an IC_50_ of 4.61 nM. However, its pharmacokinetics and related mechanisms underlying the analgesic effect remain unknown. Herein, pharmacokinetics and multiscale pharmacokinetic modelling in animals were subjected systematically to mechanistic assessment for αO-conotoxin GeXIVA[1,2]. The intramuscular bioavailability in rats and dogs was 11.47% and 13.37%, respectively. The plasma exposure of GeXIVA[1,2] increased proportionally with the experimental dose. The plasma protein binding of GeXIVA[1,2] differed between the tested animal species. The one-compartment model with the first-order absorption population pharmacokinetics model predicted doses for humans with bodyweight as the covariant. The pharmacokinetics-pharmacodynamics relationships were characterized using an inhibitory loss indirect response model with an effect compartment. Model simulations have provided potential mechanistic insights into the analgesic effects of GeXIVA[1,2] by inhibiting certain endogenous substances, which may be a key biomarker. This report is the first concerning the pharmacokinetics of GeXIVA[1,2] and its potential analgesic mechanisms based on a top-down modelling approach.

## 1. Introduction

Neuropathic pain affects approximately 7% to 10% of the general population, and the prevalence may increase with ageing of the global population [1]. The current first-line treatments for neuropathic pain comprise anticonvulsants, antidepressants, and serotonin-noradrenalin-reuptake inhibitors [2], which usually produce central side effects [2,3]. Conventional opioid analgesics lack analgesic potency for neuropathic pain, and their usage is restricted by tolerance and addiction [4]. Therefore, novel targets for neuropathic pain must be developed.

Neuronal nicotinic acetylcholine receptors (nAChRs) are crucial ligand-gated ion channels that are broadly distributed throughout the central and peripheral nervous systems [5]. α9α10 nAChR-selective antagonists can alleviate pain efficiently in various rodent neuropathic pain models [6,7]. However, only a few selective high-affinity antagonists are available for human α9α10 nAChR subtypes, and there is no commercially available therapeutic agent [8].

Conotoxins derived from the venom of *Conus* snails can selectively and efficiently modulate ion channels and provide an ideal resource for neuropharmacological tools and drug candidates screening [9]. Ziconotide (ω-conotoxin MVIIA; Prialt), a conotoxin identified from the venom of *Conus magus*, was first approved by the Food and Drug Administration in 2004 as a first-line intrathecal monotherapy for localized and diffuse chronic pain of cancer-related and non-cancer-related aetiologies [10]. However, the N-type calcium ion channel, the target of ziconotide, is in the central nervous system and requires intrathecal injection, which is troublesome in clinical administration.

αO-conotoxin GeXIVA is a novel peptide identified by our laboratory from the transcriptome of *Conus generalis*, which is native to the South China Sea. This peptide consists of 28 amino acids with 4 Cys residues and has three possible disulfide bond arrangements or isomers, i.e., Cys2–Cys20, Cys9–Cys27 (globular, GeXIVA[1,3]); Cys2–Cys27, Cys9–Cys20 (ribbon, GeXIVA[1,4]); and Cys2–Cys9, Cys20–Cys27 (bead, GeXIVA[1,2]) [11,12]. Among them, GeXIVA[1,2] (Figure 1) is the most potent selective antagonist of both rat and human α9α10 nAChRs (IC_50_ = 4.61 nM at rat α9α10 nAChR and 20.3 nM at human α9α10 nAChR) [13]. In contrast to ziconotide, the intramuscular (IM) injection of GeXIVA[1,2] is more convenient [11,12]. Our studies have indicated that GeXIVA[1,2] significantly relieves allodynia in paclitaxel [14], oxaliplatin [15], and chronic constriction nerve injury-induced rat neuropathy models and produces a cumulative analgesic effect [11]. GeXIVA[1,2] has the advantages of being nontoxic, not causing addiction and having no effect on motor function in rats [16]. These advantages make GeXIVA[1,2] a promising new analgesic therapeutic agent for clinical use.

Although the mechanism of action of GeXIVA[1,2] is generally understood, the quantitative impact of underlying key determinants influencing the rate and extent of in vivo activity remains poorly understood. Therefore, the study of pharmacokinetics (PK) in experimental animals and mechanism-based PK models that integrate key drug-specific and system-specific parameters into a quantitative framework is invaluable in understanding the potential mechanisms.

We adopted a stepwise approach to develop a multiscale, mechanistic PK model to quantitatively describe the activities of GeXIVA[1,2] in in vivo preclinical models. Such models can then (1) predict doses and PK profiles in humans using a preclinical population pharmacokinetics (PopPK) model, (2) suggest the PK-pharmacodynamics (PD) model and potential mechanism via a mechanistic PK-PD model, and (3) enable effective preclinical-to-clinical translation. This report on the PK of GeXIVA[1,2] is the first concerning IM injection and presents mechanism-based top-down modelling for elaboration of the potential analgesic mechanisms.

## 2. Materials and Methods

### 2.1. Chemicals and Reagents

GeXIVA[1,2] (TCRSSGRYCRSPYDRRRRYCRRITDACV; MW = 3451.3 Da; purity: 97.67%; Lot: P483937-2-Y) was provided by GL Biochem Ltd. (Shanghai, China). The bovine albumin (BSA), casein, Rapid Equilibrium Dialysis (RED) Device Inserts, 8K MWCO, and Baseplate were purchased from Thermo Fisher (San Jose, CA, USA). The cOmplete^TM^ proteinase inhibitor was purchased from Roche (Basel, Switzerland). Phosphate buffered saline (PBS), PBS with 0.05% Tween 20 (PBST), 3,3′,5,5′-Tetramethylbenzidine (TMB), and stop solution were purchased from Solarbio (Beijing, China). The GeXIVA[1,2]-specific antibody 4B2 and biotin-2# were prepared in our laboratory. Streptavidin-Horseradish Peroxidase was purchased from Jackson ImmunoResearch (West Grove, PA, USA)

### 2.2. Experimental Animals

Male and female beagle dogs (8–12 kg) were purchased from Beijing Rixing Ltd. (Beijing, China). Sprague–Dawley (SD) rats of both sexes, weighing 240–260 g, were obtained from Charles River (Beijing, China). Animal studies were reported in compliance with the ARRIVE guidelines [17]. All the animal procedures were approved by the Ethics Committee and Institutional Animal Care and Use Committee of the Beijing Institute of Pharmacology and Toxicology, Beijing, China (Permit Number: IACUC-DWZX-2020-698). The animals were maintained under a 12-h light/dark cycle in a temperature (25 ± 1 °C) and humidity (55 ± 5%) controlled environment. Food and water were available ad libitum.

### 2.3. PK Studies in Animals

The doses and route of administration for PK studies were chosen according to previous PD studies in which 0.11, 0.22, and 0.44 mg·kg^−1^ GeXIVA[1,2] were administered to rats by IM injection [15]. However, because of the sensitivity limitations of the detection method, these doses in rat PK studies were doubled to 0.22, 0.44, and 0.88 mg·kg^−1^. The doses in dog PK studies were converted from those in rat PD studies using the body surface area (BSA) method [18]: 0.035, 0.07, 0.14, and 0.28 mg·kg^−1^ in dogs corresponded to 0.11, 0.22, 0.44, and 0.88 mg·kg^−1^ in rats. Lyophilized powder of GeXIVA[1,2] was stored at −20 °C until used. GeXIVA[1,2] in saline was prepared fresh before each injection. A concentrated stock solution of protease inhibitor cocktail was made by using Roche cOmplete^TM^ protease inhibitor mixture tablets (1 tablet dissolved in 2 mL distilled water, 25 × concentrated). All samples and buffers were supplemented with 1× protease inhibitor cocktail (stock solution of protease inhibitor cocktail was added in buffers or samples at 4 μL/100 μL). The blood samples were immediately centrifuged (2500× *g*, 10 min, 4 °C).

#### 2.3.1. PK Studies in Rats

Twenty-four rats were randomly assigned to four groups (three male and three female rats per group) to receive a single IM dose of GeXIVA[1,2] at 0.22, 0.44, and 0.88 mg·kg^−1^ (via the left hindlimb) or a single intravenous (IV) dose at 0.44 mg·kg^−1^ (via the tail vein). Serial blood samples were collected according to Table 1.

#### 2.3.2. PK Studies in Beagle Dogs

Given the short half-life of GeXIVA[1,2], the PK data were derived from five experiments on six beagle dogs (three male and three female dogs). Dogs received a single IM dose of 0.035, 0.07, 0.14, or 0.28 mg·kg^−1^ GeXIVA[1,2] and an IV dose of 0.07 mg·kg^−1^ each time, and at least one-week intervals were allowed between each dose. Serial blood samples were collected according to Table 1.

### 2.4. Plasma Protein Binding Assay

The Rapid Equilibrium Dialysis (RED) assay [19] was used to detect the plasma protein binding rate of GeXIVA[1,2] at three concentration levels (100, 200, and 400 ng·mL^−1^) in pooled plasma of humans, beagle dogs, and rats. After 4 h of incubation at 37 °C, the sample concentration in each sample cage was detected.

### 2.5. Bioanalytical Assays

The concentration of GeXIVA[1,2] in plasma was measured using sandwich ELISA. The method was previously validated and reported [20]. A 96-well ELISA plate was first coated overnight with 1 μg per well of capture antibody 4B2 in sodium bicarbonate buffer (pH = 9.6). The plates were then blocked with 200 μL 0.5% casein in PBS for 1.5 h at 37 °C. The diluted plasma samples were added to the plate at 100 μL per well and the plate was incubated on a table concentrator for 1 h at 25 °C with shaking at 200 rpm. The plates were then washed three times with PBST and incubated with 100 μL biotinylated antibody 2# (2 μg·mL^−1^, diluted in 3% BSA PBST) for 1 h at 37 °C with shaking at 200 rpm. After washing three times with PBST, the plates were incubated with 100 μL of Streptavidin-Horseradish Peroxidase conjugate (diluted 1:10,000 in 3% BSA in PBST) for 1 h at 37 °C with shaking at 200 rpm. The plates were washed three times with PBST and developed with TMB for 10 min. The reaction was stopped by adding 50 μL of stop solution and analyzed on a Molecular Devices spectrophotometrically at a 450 nm wavelength. The relation between OD and GeXIVA[1,2] concentration was determined by a four-parameter logistic log function

### 2.6. Mechanism-Based PK Modelling

#### 2.6.1. Population PK Modelling and Human PK Prediction

The population PK (PopPK) analysis was performed using a nonlinear mixed-effects model (NLME) approach based on the PK profiles of dogs and rats after IM administration performed in this study. The first-order conditional estimation method with extended least squares (FOCE ELS) estimation was used for the PopPK model development.

##### Structural Model

One- or two-compartment models with first-order absorption were tested to determine the structural base model. Model selection was based on statistical significance among models using the minus two times the log likelihood (−2LL), Akaike information criterion (AIC), and goodness-of-fit plots (GoF). The clearance (CL), volume of distribution (V), and absorption rate constant (Ka) of GeXIVA[1,2] were estimated. Parameters obtained from the classic compartmental models were used as the initial estimates.

##### Statistical Model

Interindividual variability in PK parameters of GeXIVA[1,2] was explained using exponential error models as shown in the following equation:P_ij_ = P_tv_·exp (ŋ_ij_)(1)
where P_ij_ is the jth PK parameter estimation of the ith individual, P_tv_ is the population typical value of the jth parameter, and ŋ_ij_ is a random variable distributed with a mean of zero and a variance of ω^2^.

Residual variability was evaluated by comparing the following models:Cobs = Cpred + ε
Cobs = Cpred × (1 + ε)
Cobs = Cpred × exp (ε)
where Cobs and Cpred are the observed and predicted concentrations, respectively, and ε is a random variable distributed with a mean of zero and variances of σ^2^.

##### Covariate Model

Bodyweight (WT) was used as a covariate. The effect of WT on the PK of GeXIVA[1,2] was analysed using a stepwise method. WT was introduced to the base model when corresponding to a decrease in the OFV greater than 3.84 (*p* < 0.05) in the forward addition procedure and greater than 6.63 (*p* < 0.01) through the backwards elimination process.

##### Model Evaluation and Validation

The final established models were evaluated and verified through GoF plots, bootstrapping methods, and visual predictive checks (VPCs).

GoF plots were used to evaluate the adequacy of fitting as follows: (a) observed (DV) versus population predicted concentrations (PRED); (b) DV versus individual predicted concentrations (IPRED); (c) conditional weighted residuals (CWRES) versus PRED; and (d) quantile-quantile plot of components of CWRES.

The stability of the final model was assessed using the bootstrapping method. One thousand repeated random samplings from the original data were generated. The median and 95% CI of the parameters obtained from the bootstrap analysis were compared with the estimates of the final model.

VPCs of the final model were performed using the VPC option of Phoenix software. Time–DV concentration data were graphically superimposed on the median values and the 5th and 95th percentiles of the simulated concentration-time profiles. Model precision was expected if the DV concentration data were approximately distributed between the 95th and 5th prediction intervals (PI).

##### Model-Based Simulations

The final population PK model for GeXIVA[1,2] was used to perform simulations to predict the human PK profile. Monte Carlo simulations were used to simulate GeXIVA[1,2] exposure for 1000 virtual healthy humans weighing 70 kg. The efficacious dose was defined as the dose achieving AUC_0-inf_ in 50% of the human population, consistent with AUC_0-inf_ in rats in PD studies, and was determined using the median.

#### 2.6.2. PK-PD Modelling

A detailed PD study was published earlier by our laboratory, which reported GeXIVA[1,2] as an analgesic in the oxaliplatin-induced neuropathic pain model [17]. Oxaliplatin, a platinum-based chemotherapeutic agent, frequently causes severe neuropathic pain typically encompassing mechanical allodynia [21]. In animal models, the paw withdrawal reflex and escape response have been used extensively as behavioural assays for pain [22]. Mechanical allodynia is commonly assessed by the threshold required to evoke a motor response, typically the withdrawal of a rear paw upon plantar stimulation [23]. The manual von Frey test evaluates mechanical allodynia in mice and rats. Mechanical allodynia occurs, as evidenced by a decreased hind paw withdrawal threshold (PWT) during von Frey hair stimulation [24]. The reversed PWT reduction in the pain model usually indicates allodynia relief.

In the PD study, rats receiving oxaliplatin developed neuropathic pain between 4 and 7 days postinjection. A single IM injection of GeXIVA[1,2] at doses of 0.11, 0.22, and 0.44 mg·kg^−1^ (*n* = 12 or 13) produced a dose-dependent increase in the PWT starting at 2 h postinjection and persisting to 4 h postinjection.

All PK-PD analyses were conducted with a Phoenix model. An indirect response (IDR) model with an effect compartment was used to link the concentrations of GeXIVA[1,2] to the allodynia relief effect (PWT). An assumption was made during the modelling that GeXIVA[1,2] has the same PK profile in both normal rats and oxaliplatin-induced neuropathic pain model rats. The PK-PD modelling and validation were executed in four steps based on rat data.

Step 1: A PK model was fitted using the mean plasma concentrations after a single IM administration of 0.44 mg·kg^−1^. One-compartment models with first-order absorption were used to characterize the plasma concentration-time profiles of total GeXIVA[1,2] in rats. The equations for the model are listed below:$$\frac{d\left(\mathrm{Aa}\right)}{\mathrm{dt}}=-\mathrm{Ka}\times \mathrm{Aa}$$
$$\frac{d\left(A1\right)}{\mathrm{dt}}=\mathrm{Ka}\times \mathrm{Aa}-\mathrm{CL}\times C$$
$$C=\frac{A1}{V}$$
where Aa is the drug amount at the absorption site, Ka is the absorption rate constant, C is the drug concentration in the plasma, and V is the distribution volume.

Step 2: The mean PWT-time profiles at each dose level from the PD study were pooled (summarized in Appendix A).

Step 3: Because of the observed hysteresis between PK and PD, both the effect compartment model and IDR model were tested to establish the link between the plasma concentration and analgesic effect. The effect compartment model failed to fit the concentrations of GeXIVA[1,2] to the PWT after a single IM administration of 0.44 mg·kg^−1^, whereas the IDR model did. Using an inhibitory-loss IDR model, the time-course of the PWT was captured well. The equation for the model is listed below:$$\frac{dE}{dt}=K_{in}-K_{out}\cdot \left(1-I_{max}\cdot \frac{C}{C+IC_{50}}\right)\cdot E$$
where *dE/dt* is the rate of change in the response over time, *K_in_* represents the zero-order rate constant for the formation of the response, *K_out_* is the first-order rate constant for the loss of the response, *C* is the drug concentration in the plasma, *I_max_* is the maximal inhibition level associated with GeXIVA[1,2], and *IC_50_* is the concentration achieving 50% of *I_max_*.

Step 4: PK-PD models derived from the data at dose of 0.44 mg·kg^−1^ in Step 3 were used to project the PD profiles at doses of 0.11 and 0.22 mg·kg^−1^. The PK-PD model parameters in Step 3 were fixed, and the PWT-time profiles were simulated at 0.11, 0.22, and 0.44 mg·kg^−1^. The PWT-time profiles at 0.11 and 0.22 mg·kg^−1^ simulated by the IDR model showed significant bias compared with the observations; when the effect compartment was added, the simulations matched the observations well. The improved equations for the model based on Step 3 are listed below:$$\frac{dE}{dt}=K_{in}-K_{out}\cdot \left(1-I_{max}\cdot \frac{C_{e}}{C_{e}+IC_{50}}\right)\cdot E$$
$$\frac{dC_{e}}{dt}=K_{e0}\cdot \left(C-C_{e}\right)$$
where *Ce* is the effect compartment concentration and *K_e0_* is the distribution rate constant to the effect side.

### 2.7. Data and Statistical Analysis

PK parameters from concentrate-time data in the plasma were determined using noncompartmental analysis (NCA) with Phoenix software (Version 8.0; Pharsight, CA, USA). The dose proportionality of the PK parameter, including AUC_0-inf_ and C_max_ after IM dosing in rats and dogs, was evaluated by model-derived β values using a power model (PK = α × Dose^β^), and the critical intervals for slope (β) were 0.84–1.16. The power model analysis, PopPK, and PK-PD modelling were performed using Phoenix NLME. The above calculations to obtain the statistical calculation of individual values and Student’s t test between two groups were completed using Microsoft Excel 2016. Bioavailability (F) was calculated using the formula F = (AUC_0-inf, IM_/AUC_0-inf, IV_) × (Dose _IV_/Dose _IM_).

## 3. Results

### 3.1. PK Studies

The GeXIVA[1,2] plasma concentrations (mean ± SD) against time are shown in Figure 2. The PK parameters are summarized in Table 2.

After IM administration, GeXIVA[1,2] was rapidly absorbed in rats and dogs. The dose linearity was assessed as inconclusive both in rats and dogs according to the power model. The slope estimates (and 90% confidence interval [CI]) for parameters in the rats were as follows: area under the curve (AUC)_0-inf_, 1.26 (0.59, 1.95); and peak concentration (C_max_), 0.85 (0.061, 1.65). Those in dogs were as follows: AUC_0-inf_, 1.00 (0.65, 1.35); and C_max_, 0.75 (0.14, 1.35). Plasma exposure increased approximately in proportion to the dose according to the linear regression against dose (Appendix A). The absolute bioavailability (F) of GeXIVA[1,2] was 11.47% for rats and 13.37% for dogs. GeXIVA[1,2] is hydrophilic, which may limit its absorption after IM injection [25]. At the same time, proteolytic enzymes are ubiquitous throughout the body including the injection site, which may contribute to the presystemic degradation of GeXIVA[1,2] [26]. These two effects might be responsible for the low F of GeXIVA[1,2] after IM injection. The apparent volume of distribution during the terminal elimination phase (Vz) of GeXIVA[1,2] in rats was approximately 2-fold the volume of total body water, and the value of Vz in dogs was 0.6-fold the volume of total body water [27], indicating that the compound was less extensively distributed. The CL in both species is much higher than hepatic blood flow (rat: 13.8 mL·min^−1^, dog: 309 mL·min^−1^) [27], which points towards extra-hepatic elimination. The terminal half-life (t_1/2_) of GeXIVA[1,2] in both rats and dogs is extremely short compared with the analgesic effect of up to 6 h; this finding was consistent with its poor stability in human serum, rat plasma and dog plasma [20,28]. Thus, other mechanisms may be involved in the long-acting analgesic effect

### 3.2. Plasma Protein Binding Rate

The binding percentages of GeXIVA[1,2] to plasma proteins from human, rat, and dog plasma were 87.64 ± 7.49%, 69.81 ± 4.11%, and 55.85 ± 5.10%, respectively. Significant differences were observed among the three species (*p* < 0.05).

### 3.3. Multiscale PK Modelling

#### 3.3.1. Population PK Modelling for Human Dose and PK Prediction

A one-compartment model with first-order absorption and elimination was sufficient to characterize the PK of GeXIVA[1,2]. The interindividual variability and residual variability were described by the exponential model and log-additive error model, respectively. The final model contained WT as a significant covariate for V and CL. Compared with the basic model, the objective function value (OFV) in the final model decreased by 264.33, indicating that the incorporated covariate WT contributed to the model improvement. A summary of the model development steps is shown in Table 3.

The population parameter estimates (including Ka, V, CL, and the interindividual variability and residual variability) of the final model are presented in Table 4.

The final model of GeXIVA[1,2] reflecting the effects of covariates was described as follows:Ka = Ka_tv_·exp (ŋ_Ka_)
V = V_tv_ ·(WT/5)^dVdwt^·exp (ŋ_V_)
CL = CL_tv_·(WT/5)^dCLdwt^·exp (ŋ_Cl_)

The population estimates of CL_tv_, V_tv_, and Ka_tv_ were 113.71 L/h, 15.87 L, and 7.55 h^−1^, respectively. The relative standard error (RSE%) was 3.75–47.46% in the final model. The Eta shrinkage values of the estimated PK parameters were considered acceptable (4.46–25.25%) (Table 4).

Goodness-of-fit plots of the final models for GeXIVA[1,2] are shown in Figure 3A–D. The observed and predicted concentrations of GeXIVA[1,2] were very consistent in the final model. The CWRES were well distributed around zero. In the bootstrapping for the final model, all 1000 replications were run successfully. The estimated parameters in the final model were close to the median values in the bootstrapping analysis and fell within the 95% CIs (Table 4). In the VPC for the final model, most of these observed data were distributed within the 90% PI of the predicted value (Figure 3E). The final established model should be reliable and robust.

The final PopPK models were used to simulate the GeXIVA[1,2] concentration-time profiles for healthy humans weighing 70 kg (*n* = 1000). The plasma exposures (AUC_0-inf_) of GeXIVA[1,2] in humans at doses of 11.92 and 30.85 μg·kg^−1^ (109.56 and 283.42 ng·min·mL^−1^, respectively) were consistent with the exposures in rats at doses of 0.22 and 0.44 mg·kg^−1^ (109.84 and 283.52 ng·min·mL^−1^, respectively). Thus, doses ranging from 11.92 to 30.85 μg·kg^−1^ may be efficacious for humans. The CL/F, Vz/F, t_1/2_, C_max_, and T_max_ were 108.87 mL·min^−1^·kg^−1^, 4244.28 mL·kg^−1^, 27.02 min, 1.90 ng·mL^−1^, and 18 min and 108.87 mL·min^−1^·kg^−1^, 4244.30 mL·kg^−1^, 27.02 min, 4.90 ng·mL^−1^, and 18 min for doses of 11.92 μg·kg^−1^ and 30.85 μg·kg^−1^, respectively.

#### 3.3.2. Mechanistic PK-PD Modelling

The time course of PWT behaviour reflected the analgesic effects observed after GeXIVA[1,2] administration. A one-compartment model with first-order absorption and elimination derived from the PK profiles after the IM administration of 0.44 mg·kg^−1^ GeXIVA[1,2] was sufficient to characterize the PK profile of the rats after the administration of 0.22, 0.44, and 0.88 mg·kg^−1^ GeXIVA[1,2] (Figure 4A,B). Next, the PK parameters were fixed, and the observed PWT values were used as the input to the PD model using the Phoenix model. When plotting the PWT value versus the plasma concentration of GeXIVA[1,2], hysteresis was observed at all doses in rats. The inhibitory loss IDR model with an effect compartment described the PK-PD relationships (Figure 4C,D).

The parameter estimates were as follows: Ka, 68.19 h^−1^; V, 12.83 L·kg^−1^; CL, 86.22 L·h^−1^·kg^−1^; Ke0, 1.12 h^−1^; K_in_, 1.21 h^−1^; K_out_, 0.53; I_max_, 2.16; and IC_50_, 0.16 ng·mL^−1^. The PWT values at 1, 2, 4, and 6 h postdosing were simulated, and the AUC of the PWT was calculated (Appendix A).

The observed PWT values and the predicted PWT values were matched well (Figure 4E). The ratios of the predicted-to-observed PWT values were within 0.69–1.22-fold range, and the AUC of the effect was within 1.03–1.11-fold range for all three doses, indicating that the PK-PD model could adequately describe the PK-PD relationships (Figure 4E and Appendix A). The linear regression of the predicted PWT values from the PK-PD models plotted against the observed experimental data showed good correlation between the observed and predicted data (Figure 4D).

### 3.4. Potential Mechanism of GeXIVA[1,2] Based on Multiscale PK Modelling

Model simulations suggested that the effect compartment was added to link the PK and PD for a better fit, implying a lag in effectiveness [29,30]. Both the rat and dog PK experiments showed that GeXIVA[1,2] has a short half-life in animals; however, the analgesic effect of GeXIVA[1,2] started at 2 h postinjection and persisted to 4 h postinjection in rats. Model simulations provided potential mechanistic insights that the response to GeXIVA[1,2] may result from inhibition of the factors causing downstream signalling of pain [31]. GeXIVA[1,2] in plasma translocated to the effect response with a delay. After binding to α9α10 nAChR, downstream signalling pathways were affected, possibly exerting an analgesic effect by inhibiting certain endogenous factors (Figure 5). The key endogenous factors may be crucial PD biomarkers. We believe that this observation provides a valuable contribution to increasing the understanding of the mechanisms of GeXIVA[1,2]. Next, we sought to identify and confirm the endogenous PD biomarker.

## 4. Discussion

The conotoxin family comprises disulfide-rich peptides derived from cone snails, which efficiently and selectively target ion channels. Conotoxins targeting specific targets have the potential to be developed into analgesics [32].

αO-conotoxin GeXIVA[1,2] is a structurally novel peptide [13]. GeXIVA[1,2] targeted α9α10 nAChR and provided both long- and short-term analgesic effects when administered by IM injection. Our laboratory owns the independent intellectual property rights of GeXIVA and its mutant, which have been patented in China, the United States, Japan, and Europe (PCT/CN2013/076967, PCT/CN2020/090978). GeXIVA[1,2], a novel nonaddictive analgesic, is valuable for clinical development.

Despite this success, the quantitative impact of key drug-specific and system-specific determinants associated with GeXIVA[1,2] activity is not well understood. Establishing PopPK and PK-PD relationships for GeXIVA[1,2] is challenging, and no established paradigms or guidelines exist to predict safe and efficacious dose levels for GeXIVA[1,2] in humans. The development of multiscale system PK models could be a highly beneficial first step to identifying key determinants associated with the kinetics and activities of these agents. The characterization of the PopPK and PK-PD relationships of GeXIVA[1,2] presents many challenges and unique opportunities.

In the present study, in vivo PK studies of GeXIVA[1,2] were first reported. GeXIVA[1,2] had a shorter t_1/2_ in both rats and dogs. The t_1/2_ value after IM administration was prolonged when the dose exceeded 0.44 mg·kg^−1^ in rats and 0.14 mg·kg^−1^ in dogs, a finding that is consistent with the decrease in the CL as the dose increases, likely because of target-mediated drug disposition (TMDD) [26]. The plasma exposure increased approximately in proportion to the dose. Therefore, GeXIVA[1,2] might show linear PK at lower doses. The linear dose ranged from 0.22 to 0.44 mg·kg^−1^ for rats and from 0.035 to 0.14 mg·kg^−1^ for dogs. When the dose exceeds these ranges, nonlinear PK may be observed. Overall, the t_1/2_ of GeXIVA[1,2] was much shorter than its analgesic effect. This finding suggests that GeXIVA[1,2] exerts its analgesic effects through complex mechanisms. Fortunately, the analgesic effect of GeXIVA[1,2] is maintained for about 6 h. If the peptide could be further protected (i.e., engineered), such an effect would probably be further extended. A number of strategies have been developed to augment peptide activity, lengthen their half-life, and improve their distribution [33]. Cyclization and PEGylation of other conotoxins have been carried out in our laboratory, both of which significantly improved the stability of the conotoxins [34,35]. These works also provide experience for the modification of GeXIVA[1,2].

The PopPK approach can predict the human dose and PK profile from preclinical species [36,37]. To determine the factors that might influence the variability of parameters, various error models were assessed. In the current analysis, the power exponents of the body weight effect on CL and V were 0.52 and 1.00, respectively. Bodyweight was not a significant covariate of Ka, suggesting that Ka might be similar among species. The VPC simulations and bootstrap replicates demonstrated that the final models were adequately stable and accurate. Based on the PopPK final model, an efficacious dose of 11.92–30.85 μg·kg^−1^ in 70 kg healthy humans was predicted. Cancer patients undergoing chemotherapy are often underweight and have associated organ failure. The current model can be further used to cope with different requirements (e.g., individualization of dosing). Our results are expected to be useful as the basis for the clinical use of GeXIVA[1,2].

In the present study, an IDR model combined with an effect compartment was a better model to describe the relationship between the GeXIVA[1,2] concentration in plasma and drug effect. Von Frey filaments are the gold standard for evaluating sensory thresholds in rats. Mechanical sensitivity is expressed by the PWT after pricking the hind limb with von Frey filaments [38]. The predicted PWT of all three doses showed a good fit to the observed value. According to the model, GeXIVA[1,2] in plasma translocates to the effect response with delay. After the binding of GeXIVA[1,2] to α9α10 nAChR, downstream signalling pathways were affected, possibly leading to an analgesic effect by inhibiting certain endogenous factors. The inhibitory loss model indicated that GeXIVA[1,2] might inhibit the generation of those pain-causing factors. Considering the close relationship between α9α10 nAChRs and the immune response [39], certain inflammatory pain-causing cytokines, such as TNF-α and IL-2, may be key factors in the analgesic effect of GeXIVA[1,2] [40]. The key endogenous factors might be important PD biomarkers.

Flores-Murrieta et al. characterized the PK-PD relationship of tolmetin in a rat inflammatory pain model using an IDR model [41]. The validity of the PK-PD model in reflecting the mechanism of action of nonsteroidal anti-inflammatory drugs was confirmed through a close correlation between the drug concentration and value in an in vitro prostaglandin synthesis inhibition assay. Prostaglandins are key biomarkers. These examples of PK-PD modelling provide improved insights into the mechanism of action of analgesics. Translational PK-PD modelling will not eliminate the current issues with preclinical models but has value as a tool for increasing the success in selecting the right drugs for the right pharmacological targets during early clinical development. The selection of the drug targets starts with gaining a better understanding of the relationship between exposure and target binding and subsequent quantitative correlation with a downstream effect following target activation. The greatest challenges in translational pain research from animals to humans arise from the lack of availability of predictive animal pain models for human pain conditions because of the disconnect between animal and human pain aetiologies and differences in pain pathways between animals and humans [42]. The translation of PK-PD data obtained from animal pain behaviour studies to patient outcome measures such as pain relief remains limited. Various challenges must be overcome to fill the animal-to-patient translational gap in pain research. Our proposed mechanism suggests the need to confirm key downstream endogenous substances as PD biomarkers and develop a more mechanistic PK-PD model.

## 5. Conclusions

This work represents the first report of the PK of a novel, nonaddictive, conotoxin-derived therapeutic agent administered through IM injection. Herein, we describe a mechanism-based model developed to characterize PopPK and PK-PD of GeXIVA[1,2]. Although a further evolved model incorporating other pertinent components will be necessary to describe the in vivo behaviour of GeXIVA[1,2], our multiscale translational PK model integrated drug- and system-specific parameters and could characterize GeXIVA[1,2] activity in vivo. Model simulations provided potential mechanistic insights, showing that after GeXIVA[1,2] bound to α9α10 nAChRs, downstream signalling pathways were affected with delay, possibly resulting in an analgesic effect by inhibiting certain endogenous factors, which could be key PD biomarkers.

## Figures and Tables

**Figure 1 pharmaceutics-14-01789-f001:**
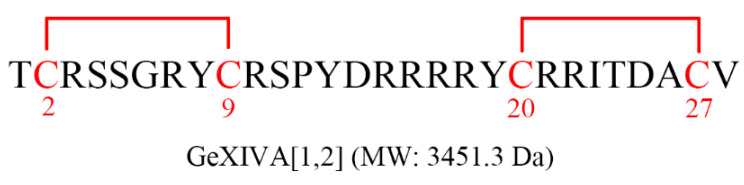
Amino acid sequence of GeXIVA[1,2]. GeXIVA[1,2] is a 28-amino acid peptide. It includes 4 cysteines with disulfide bonds between Cys2–Cys9 and Cys20–Cys27. Red lines indicate the disulfide bonds.

**Figure 2 pharmaceutics-14-01789-f002:**
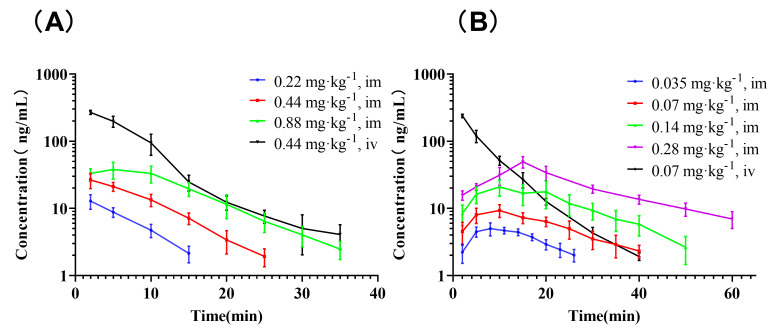
Mean plasma concentrations of GeXIVA[1,2] over time in rats (A) and dogs (B) after intramuscular (IM) or intravenous (IV) administration of GeXIVA[1,2] (*n* = 6).

**Figure 3 pharmaceutics-14-01789-f003:**
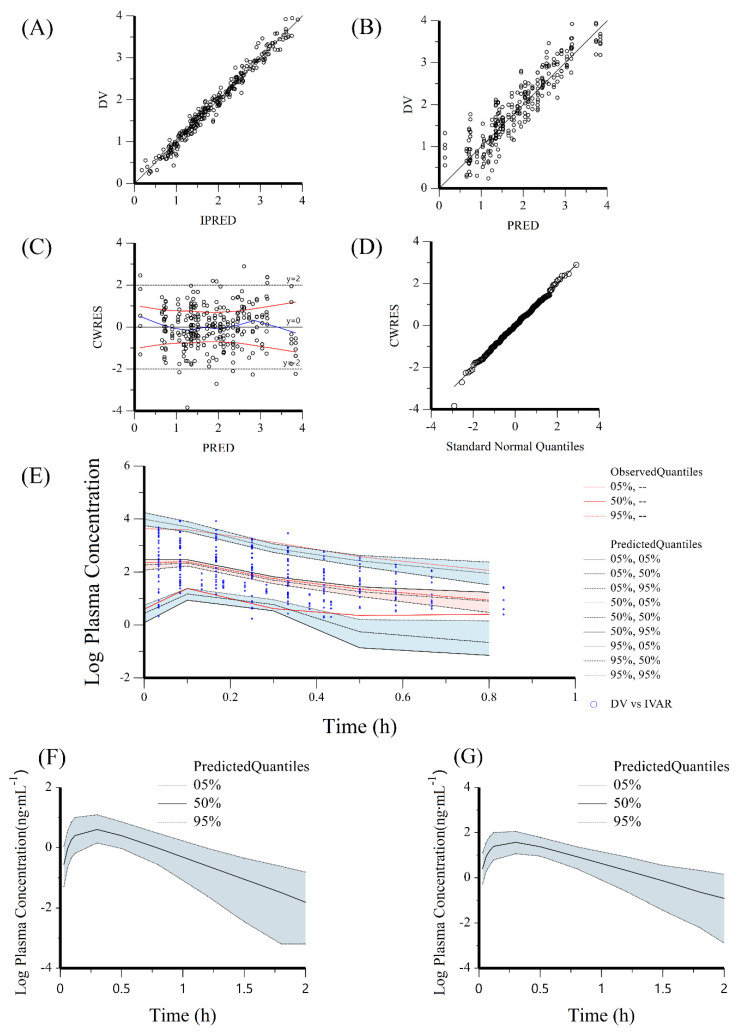
Validation of the final model based on observed data and simulated PK profile based on the final model in 70 kg healthy humans. (**A**) DV against IPRED. (**B**) DV against PRED. (**C**) CWRES against PRED. (**D**) Quantile-quantile plot of the components of the CWERS. (**E**) Visual predictive check. (**F**) The mean and prediction interval results from the visual predictive check versus time after IM administration of GeXIVA[1,2] at the dose of 11.92 μg·kg^−1^ in 70 kg healthy humans. (**G**) The mean and prediction interval results from the visual predictive check versus time after IM administration of GeXIVA[1,2] at the dose of 30.85 μg·kg^−1^ GeXIVA[1,2] dose in 70 kg healthy humans.

**Figure 4 pharmaceutics-14-01789-f004:**
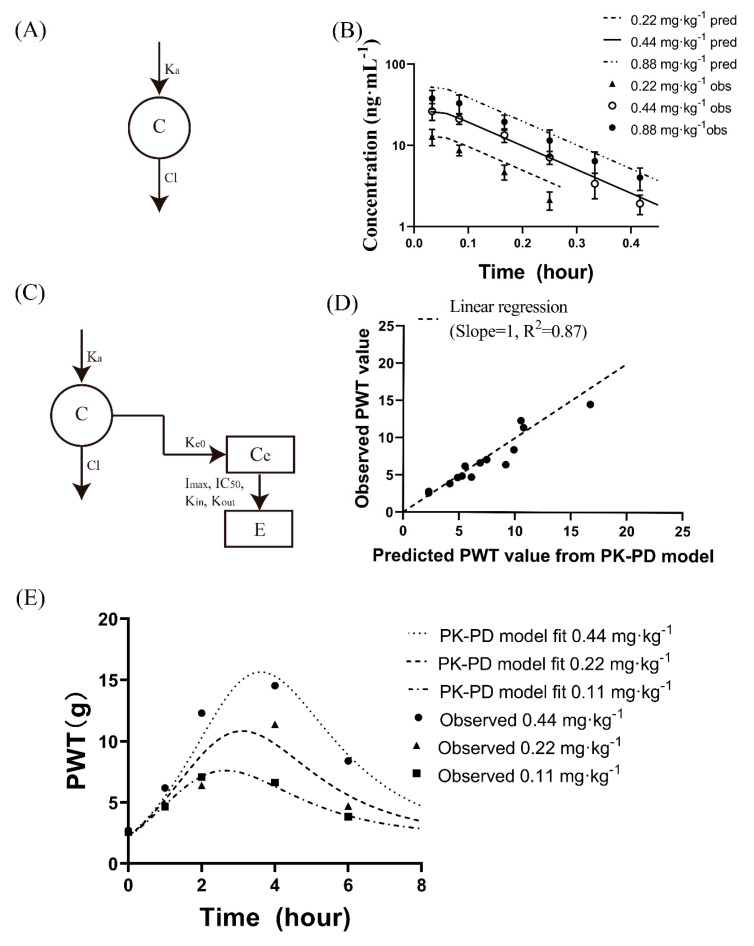
PK-PD modelling for GeXIVA[1,2]. (**A**) Schematic of the one-compartmental model used to fit the concentration–time profiles of GeXIVA[1,2] derived from the concentration–time profiles after the IM administration of 0.44 mg·kg^−1^ GeXIVA[1,2]. (**B**) Observed concentration–time profiles versus predicted concentration–time profiles of GeXIVA[1,2] after the IM administration of 0.22, 0.44, and 0.88 mg·kg^−1^ GeXIVA[1,2] in rats. (**C**) Schematic overview of the PK-PD model for GeXIVA[1,2]. (**D**) Observed PWT value in rats versus predicted PWT value from the PK-PD model. R^2^ for the linear regression (slope = 1) was 0.87. (**E**) PK-PD model fitted PWT-time profiles versus observed PWT-time profiles after the IM administration of 0.11, 0.22, and 0.44 mg·kg^−1^ GeXIVA[1,2].

**Figure 5 pharmaceutics-14-01789-f005:**
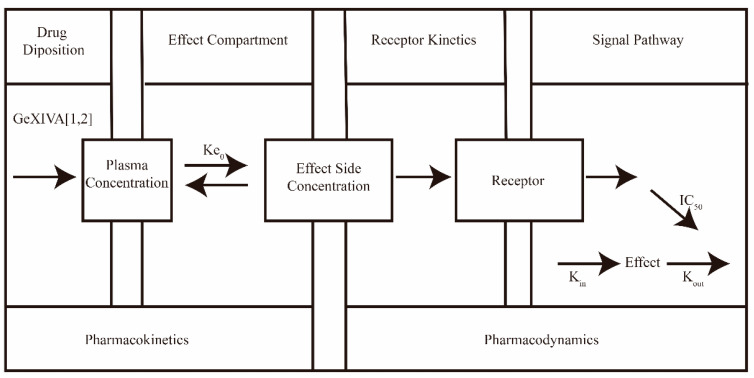
A schematic overview of the PK-PD model for GeXIVA[1,2]. GeXIVA[1,2] in plasma translocated to the effect side with the rate K_e0_. After binding to the α9α10 nAChR, downstream signalling pathways were affected, possibly resulting in an analgesic effect by inhibiting certain endogenous factors.

**Table 1 pharmaceutics-14-01789-t001:** Results of pharmacokinetic studies of GeXIVA[1,2] in rats and dogs.

Spices	Dose (mg·kg^−1^)	Weight (kg)	Number and Sex	PK Sampling Scheme (min)
Rat	IM, 0.22	0.25 (0.24–0.26)	3 M + 3 F	2, 5, 10, 15
	IM, 0.44		3 M + 3 F	2, 5, 10, 15, 20, 25
	IM, 0.88		3 M + 3 F	2, 5, 10, 15, 20, 25, 30, 35
	IV, 0.44		3 M + 3 F	2, 5, 10, 15, 20, 25, 30, 35
Dog	IM, 0.035	10 (8–12)	3 M + 3 F	2, 5, 8, 11, 14, 17, 20, 22, 26
	IM, 0.07		3 M + 3 F	2, 5, 10, 15, 20, 25, 30, 35
	IM, 0.14		3 M + 3F	2, 5, 10, 15, 20, 25, 30, 35, 40, 50
	IM, 0.28		3 M + 2 F	2, 5, 10, 15, 20, 25, 30, 40, 50, 60
	IV, 0.07		3 M + 3 F	2, 5, 10, 15, 20, 30, 40

**Table 2 pharmaceutics-14-01789-t002:** Pharmacokinetic parameters of GeXIVA[1,2] in rats and dogs after a single dose of GeXIVA[1,2] (*n* = 6).

Species	Route	Dose (mg·kg^−1^)	AUC_(0-inf)_ (ng·min·mL^−1^)	CL ^1^ (mL·min^−1^·kg^−1^)	Vz ^2^ (mL·kg^−1^)	t_1/2_ (min)	C_max_ (ng·mL^−1^)	T_max_ (min)
Rat	IV	0.44	2472.84 ± 272.29	179.81 ± 20.50	1396.15 ± 328.79	5.37 ± 0.99	269.17 ± 18.68	2.00 ± 0.00
	IM	0.22	109.84 ± 17.72	2047.91 ± 337.14	14,572.0 ± 2910.81	4.98 ± 0.91	12.86 ± 3.17	2.00 ± 0.00
	IM	0.44	283.52 ± 39.76	1578.12 ± 225.36	11,686.1 ± 1221.54	5.20 ± 0.78	27.17 ± 5.79	3.00 ± 1.55
	IM	0.88	637.17 ± 101.91	1415.13 ± 258.46	17,933.3 ± 8602.70	8.67 ± 3.70	41.65 ± 7.63	5.67 ± 3.61
Dog	IV	0.07	1943.66 ± 132.18	36.15 ± 2.48	353.53 ± 72.14	6.76 ± 1.15	238.57 ± 13.90	2.00 ± 0.00
	IM	0.035	121.23 ± 19.58	295.01 ± 47.07	4261.72 ± 538.69	10.24 ± 2.32	5.32 ± 0.88	9.50 ± 3.15
	IM	0.07	259.78 ± 59.90	281.62 ± 63.48	4968.49 ± 1021.32	12.50 ± 2.38	9.40 ± 2.00	9.17 ± 2.04
	IM	0.14	600.42 ± 172.99	251.61 ± 78.50	4170.37 ± 1411.30	11.50 ± 1.31	21.82 ± 7.09	11.67 ± 4.08
	IM	0.28	1440.13 ± 128.75	195.67 ± 17.27	5565.50 ± 1948.35	19.65 ± 6.21	49.36 ± 9.50	15.00 ± 0.00

AUC_(0-inf)_: area under the plasma concentration-time curve from 0 to infinity; CL: total plasma clearance; Vz: apparent volume of distribution during the terminal elimination phase; t_1/2_: terminal half-life; C_max_: peak concentration; T_max_: peak concentration time. ^1^ for intramuscular (IM) injection, CL/F; ^2^ for IM injection, Vz/F.

**Table 3 pharmaceutics-14-01789-t003:** Summary of the model building steps for the population PK of GeXIVA[1,2].

Model	Description	nParm	−2LL	AIC	Δ−2LL	ΔAIC
Absorption Model						
01	T_lag_	9	2195.57	2213.57		
02 ^a^	No T_lag_	7	1535.18	1549.18	−660.39	−664.39
Residual Model						
02-01	Additive	7	1535.18	1549.18		
02-02 ^a^	Log additive	7	230.50	244.50	−1304.68	−1304.68
02-03	Muliticative	7	1259.60	1273.60	1029.10	1029.10
IIV Model						
02-02-01 ^a^	Do not remove		230.50	244.50		
02-02-02	Remove Ka	6	213.50	225.50	−17.00	−19.00
02-02-03	Remove V	6	419.64	431.64	206.14	206.14
02-02-04	Remove Cl	5	501.89	513.89	82.25	82.25
02-02-05	Remove Ka and V	5	531.09	541.09	29.20	27.20
02-02-06	Remove Ka and CL	5	729.47	739.47	198.39	198.39
02-02-07	Remove V and CL	5	789.12	799.12	59.65	59.65
Covariates Model						
02-02-01-01	WT on CL	8	115.95	131.95	−114.56	−112.56
02-02-01-02	WT on V	8	75.52	91.52	−154.98	−152.98
02-02-01-03 ^b^	WT on CL and V	9	−33.82	−15.82	−264.33	−260.33

^a^, Selected model; ^b^, Final model; −2LL, minus two times the log likelihood; AIC, Akaike Information Criterion; BIC, Bayesian Information Criterion; nParm, number of parameters; IIV, Interindividual variability.

**Table 4 pharmaceutics-14-01789-t004:** Population pharmacokinetic parameters for GeXIVA[1,2] in the final model and bootstrap.

Parameter	Final Model				Bootstrap	
	Estimate	SE	RSE (%)	Shrinkage	Median	95% CI
Ka_tv_	7.55	0.48	6.35		7.54	6.54~8.50
V_tv_	15.87	1.30	8.22		15.80	12.91~18.53
CL_tv_	113.71	5.01	4.40		113.48	104.13~124.14
dCLdwt	0.52	0.02	3.75		0.52	0.48~0.56
dVdwt	1.10	0.05	4.50		1.10	1.01~1.21
ω^2^Ka	0.10	0.05	47.46	0.25	0.16	
ω^2^V	0.04	0.02	36.00	0.25		
ω^2^Cl	0.04	0.01	18.26	0.04		
σ	0.16	0.01	8.23		0.16	0.13~0.18

## Data Availability

All related data and methods are presented in this paper. Additional inquiries should be addressed to the corresponding author.

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
