# Peer review of "Novel αO-conotoxin GeXIVA[1,2] Nonaddictive Analgesic with Pharmacokinetic Modelling-Based Mechanistic Assessment"

_pharmaceutics, 2022, doi:10.3390/pharmaceutics14091789_

Round 1

Reviewer 1 Report

The manuscript pharmaceutics-1850455 may be accepted for publication if the authors justify two important aspects, possibly with the inclusion of 2 paragraphs about:

1. Stability of the peptide under physiological conditions

2. Criteria for the dose applied to animals. It is necessary to explain, in detail, the use of these values and not just refer via [1,2].

Reviewer 2 Report

Manuscript contains comprehensive animal PK and efficacy data of alphaO-conotoxin. Population PK and especially PKPD-modeling provides valuable information on compound behavior. However, there are several inconsistencies and incompletely justified extrapolations that need to be reconsidered and/or revised. Most importantly, prediction of human PK is not well justified and it is in quite minor role since actual data in manuscript is otherwise good. Consider whether this could be left out and focus on actual findings of the study. Further, species differences (human vs. rat) in primary pharmacology is not discussed/addressed. According to ref#12, human IC50 is 4-times lower than that of rat, therefore 4-times higher exposure need to be obtained in human. Please omit from the abstract text following sentence: Model simulations have provided potential… as I could not find what are those endogenous substances? Line 400-403 Is such discussion on patent status of the molecule necessary in a scientific article?

Population PK modeling

-          it is unclear whether all PK data was included in the model? or only IM?

-        Main putative elimination pathways for this kind of molecule should be briefly discussed since extrapolation of human PK is sought. Now all scaling is based on body weight. Is it expected that there are no differences in mechanisms or enzyme/transporter affinities between species? TMDD is discussed on line 418 but e.g. species differences in target affinities are not mentioned.

-          Figure 3E. Does this contain all different animals, doses, and routes somehow normalized? Y-axis unit is likely wrong since the scale is likely some log or ln? not directly ng/mL? different quantile legends are pretty difficult to read.

-          It should be clear in fig 3 legend that A-E are animal data based on actual observations and F-G are human predictions

-          Overall, really consider whether human PK predictions provide sufficient value since discussion states several uncertainties and no discussion on validity of body weight based scaling on these mechanisms. At least provide some references that possibly provide data on validity of this approach.

-          PPB is reported but it has not been used to anything in the modeling. As poor stability in plasma is stated (line 275) it should be discussed how in vitro PPB recovery values were after 4 h incubation

-          2.6.1.5 should discuss validity of direct BW scaling of CL and BW and also direct incorporation of population variability parameters from quite homogenous animal populations to likely much more variable human (patient) population

-          Human efficacious dose prediction is based on efficacy in rats. There is no discussion on species differences in primary activity or the fact that statistical difference in artificial animal pain models are not necessarily indicative of clinically significant exposure levels in humans (line 460-463). Therefore, prediction of efficacy in humans is not well justified here

-          line 313 there is an error in the exponent. Should not be dKadv for V?

PKPD modeling

-          Please provide all differential equations. e.g. line 218 equation for plasma concentration appears erroneous

-          Why was only one PK dose level used while (apparently) all were used for PopPK approach?

-          line 352 describes hysteresis plot but this is not shown

-          please compare Km value from PKPD model to in vitro observed value. Is current in vitro assay predictive of in vivo

-           

Miscellaneous

-          generally results should be rounded to reasonable accuracy. E.g. PK parameter table has up to six significant numbers. Two to three would suffice and make the manuscript more readable

-          Consider whether statistical testing of PK parameters in in table 2 is really necessary. Dose linearity/proportionality is discussed so it is quite obvious that different dose levels have statistically significant differences in exposure

-          IM bioavailability numbers are mentioned only in abstract but not in main text. Calculation is not described. Moreover, some discussion on the reason for low bioavailibiility mechanism for this type of molecule is warranted. Is it reasonable to assume that this low BA is also expected in human

-          In vitro potency mentioned only in abstract. Should also be in introduction with literature reference

-          Abstract (p1. line 20) states human predicted doses but these are not thoroughly justified and, while discussion (line 404) states that mechanistic prediction is challenging and human PK prediction is in very minor role in actual manuscript, probably should not be raised in abstract item

-          abstract line 22-24 and line 388 discuss endogenous biomarkers but the manuscript does not describe any such. Therefore, this part of abstract does not correlate to the content of the manuscript.

-          result section 3.4 is more like discussion item than result of this study (that had not been discussed in previous sections)

-          Dose linearity/proportionality is ambiguously discussed. Line 263 states dose linearity was assessed as inconclusive in both dogs and rats. Line 267 states that plasma exposure increased approximately dose proportionally. lines 415-422 discuss non-linear CL and T1/2 but linear exposure and possibility for non-linear doses. abstract line 18 states proportionality

-          line 269-270 puts Vz values to physiological context by comparing them to body fluid volumes. Similar approach should be done for IV CL values by comparing to liver blood flow. although elimination may be extrahepatic but puts the value in context with standard pharmacokinetics. In addition, might suggest modifications to the molecule to decrease CL for improved duration of action.

Reviewer 3 Report

     This work appears as a well planned and made, easy to read, interesting and potentially used research piece. It collects a series of analysis to conform a solid and clear-cut in vivo pharmacokinetics study of the αO-conotoxin GeXIVA[1,2], a no addictive peptide analgesic targeting the α9α10 nicotinic acetylcholine receptor (nAChR), in rat and dog models, intramuscularly administered. The work generates a mechanism-based model that allows to start characterizing the activity of this αO-conotoxin in vivo, to start describing the in vivo behaviour of GeXIVA[1,2], a necessary step to convert it in a useful therapeutic agent.  Besides, the modelling allowed to predict the efficacious doses of it in healthy humans. Nevertheless, in this reviewer's opinion, there are a series of points of it that authors should consider for improvement, mostly for the benefit of readers.

-Authors use the term "Novel" to define this αO-conotoxin, although its first report was at least published in 2015 (PNAS) by the same group. Probably the present is the first time that a solid in vivo and pharmacokinetics study of it is made ... but perhaps authors should take into account that now the molecule has been at the field and bibliography for at least 7 years. Certainly, this is not a larger period for the development of a useful drug, and this molecule has a great potentiality to become it.  More important perhaps is the use of the term GeXIVA[1,2] that refers to the selection of the so called "bed" disulfide-linked isomer, the most potent and promising variant of it, by now; this terminology is quite a bit confusing for non-specialized readers, particularly when used in the main text together with bibliographic references ... because it seems to indicate that it brings to references 1 and 2. Anyway, authors should describe in each main paper, including the present one, such terminology (bead/ ribbon/ globular S-S patterns, and 1,2/ 1,3 / and 1,4 connectivities, to facilitate understanding, and save readers of the need to read previous publications.

-Given that the same authors in previous publications commented the appearance of GeXIVA as distinct disulfide-linked isomers, and the poor stability of them, the precise conditions in which this molecule is kept/stored, analysed and administered to animals must be clearly stated in this paper, as pH (a great determinant for disulfides and its scrambling), buffer, temperature, potential use of adjuvants ...etc, without referring to previous publications. Also, in the manuscript it is clearly indicated the short half-life (t 1/2) of GeXIVA in the animals (i.e. Table II and lines 272 and 379). To know in detail the reason of this short half-life, not clarified (or incorporated) now in this work, could be a useful addition in next ones (i.e. analysing the fragmentation of GeXIVA in plasma and/or in vivo by proteomic/immunologic approaches). Fortunately the generated analgesic effect is kept for longer (about 6h), but if the molecule could further protected (i.e. engineered), such effect probably would be further extended.  Both issues, stability and detailed degradation of GeXIVA would merit to be shortly commented in this work.

-Considering the important number of abbreviations, it would be useful to collect them in an abbreviations section. Noteworthy, this reviewer has not been able to detect and download the Supplementary Materials indicated in lines 480-481, neither together with the manuscript to download, nor within the same main manuscript.  

Round 2

Reviewer 1 Report

This reviewer is grateful for the clarification and suggests the publication of the manuscript.

Reviewer 2 Report

The authors have carefully assessed all concerns.